# Analysis of Chemical Constituents of *Chrysanthemum morifolium* Extract and Its Effect on Postprandial Lipid Metabolism in Healthy Adults

**DOI:** 10.3390/molecules28020579

**Published:** 2023-01-06

**Authors:** Lin Chen, Jihan Sun, Zhengyu Pan, Yifei Lu, Zhaodan Wang, Ligang Yang, Guiju Sun

**Affiliations:** 1Department of Nutrition and Food Hygiene, Key Laboratory of Environmental Medicine Engineering of Ministry of Education, School of Public Healthy, Southeast University, Nanjing 210009, China; 2College of Biology and Food Engineering, Technology Research Center of Characteristic Biological Resources in Northeast of Chongqing, Chongqing Three Gorges University, Chongqing 404000, China

**Keywords:** *Imperial Chrysanthemum*, UHPLC-MS, flavonoids, high-fat meal, postprandial lipemia, antioxidant status

## Abstract

Chrysanthemum extract possesses antioxidant potential and carbohydrate and fat digestive enzyme inhibitory in vitro. However, no evidence supporting chrysanthemum in modulation of postprandial lipemia and antioxidant status in humans presently exists. This study was to analyze the composition of *Imperial Chrysanthemum* (IC) extract and determine the effect on changes in postprandial glycemic and lipemic response and antioxidant status in adults after consumption of a high-fat (HF) meal. UHPLC-MS method was used to analyze the components of two kinds of IC extracts (IC-P/IC-E) and in vitro antioxidant activities were evaluated using 1,1-diphenyl-2-picrylhydraxyl (DPPH), 2,2-azino-bis-3-ethylbenzothiazoline-6-sulfonic acid (ABTS) and Hydroxyl radical (HR) radical scavenging assays. Following a randomized design, 37 healthy adults (age, 25.2 ± 2.6 years, and BMI, 20.9 ± 1.5 kg/m^2^) were assigned to two groups that consumed the HF meal, or HF meal supplemented by IC extract. Blood samples were collected at fasting state and then at 0.5, 1, 2, 4, 6 and 8 h after the meal consumption. There were 12 compounds with relative content of more than 1% of the extracts, of which amino acid and derivatives, flavonoids, carboxylic acids and derivatives were the main components. Compared with IC-E, the contents of flavonoids in IC-P increased significantly (*p* < 0.05), and the cynaroside content exceeded 30%. In addition, IC-P showed strong free radical scavenging activity against DPPH, ABTS and HR radicals. Furthermore, according to repeated–measures ANOVA, significant differences were observed in the maximal changes for postprandial glucose, TG, T-AOC and MDA among the two groups. Postprandial glucose has significant difference between the two groups at 1 h after meal and the level in IC group was significantly lower than that in control group. No significant differences were observed in the incremental area under the curve (iAUC) among the two groups. IC significantly improved the serum antioxidant status, as characterized by increased postprandial serum T-AOC, SOD, GSH and decreased MDA. This finding suggests that IC can be used as a natural ingredient for reducing postprandial lipemia and improving the antioxidant status after consuming a HF meal.

## 1. Introduction

Since ancient times, people have accumulated a rich experience and knowledge of edible and medicinal flowers in the long history of eating flowers [1]. Currently, flowers of over 180 species have been used worldwide as dietary supplements in the form of tea, wine, food ingredients, or even vegetables [2,3]. Edible flowers mostly belong to Asteraceae, Fabaceae, Liliaceae, Rosaceae and other plants, among which chrysanthemum is one of the most important [4,5,6]. Chrysanthemum originates in China and has been used for over 3000 years [7]. Chrysanthemum is often used as a soaking or boiling material to drink in China [8]. It not only has a unique appearance, aroma and taste, but its active ingredients and medicinal value have also attracted researchers’ attention [9]. As a traditional Chinese medicine, chrysanthemum can be used for “common cold with wind-heat patten”, “swelling and pain of eyes”, and it is effective in clinical practice for a wide variety of diseases such as headache, dizziness, hypertension, angina pectoris, arrhythmia, coronary heart disease, etc. [10]. With the growing demand for higher living standards and the enhancement of health awareness, edible plant medicines have recently become important in people’s daily life. As an herbal medicine, chrysanthemums have advantages of an antioxidant effect, anti-inflammation, antioxidant, antibacterial activity, anti-obesity, etc. [11,12,13]. However, these studies mainly focus on the extraction of components and functional analysis [14,15,16]. In vitro and in vivo efficacy experiments have been conducted mainly in cells and animals [17,18]. The latest review of chrysanthemum detailed the chemical composition and pharmacological effects, but did not mention hypolipidemic effects [19]. Chrysanthemum has a biological effect related to lipid metabolism, and its extract decreased the components of serum lipid profile such as triglyceride, total cholesterol and low-density lipoprotein cholesterol levels in rats [20,21]. Nevertheless, the effect of chrysanthemum extract with a high-fat (HF) meal on postprandial lipemic and antioxidant activities in humans has never been investigated.

*Dendranthema morifolium* (Ramat.) Tzvel. ‘*Imperial chrysanthemum*’ (IC) is a traditional Chinese medicinal herb of the family Compositae with dried anthodium of Ramat, and has been eaten for more than 1000 years [22]. IC has been used extensively in tea drinks and snacks. Its characteristic “one cup of tea with one flower” makes it distinctive from other scented flower teas. On the one hand, IC has a colorful appearance and pleasant fragrance; on the other, it contains a variety of natural active ingredients. At present, the research on IC mainly focuses on its extract components [23,24] and in vitro activities, such as polysaccharide composition and structure [25], antioxidant [22,26] and prebiotic properties [27], while there are few studies on flavonoids and effects on lowering blood lipids [28]. Chrysanthemum flavonoid, as a large class in numerous bioactive compounds in chrysanthemum, has a great antihyperlipidemic and antioxidation effect. The extraction, isolation and purification of flavonoid, having hypoglycemic blood lipid and antioxidation effects in chrysanthemum, are important to develop resources and develop its effective components.

Our research group has been engaged in research into chrysanthemum flavonoids for a long time and found that chrysanthemum flavonoids had a stronger antioxidant and lipid metabolism regulation ability [29], but there is still a lack of population research. This article mainly studied and discussed the chemical constituents extracted from IC and investigated the effects of postprandial lipid metabolism in healthy adults by focusing on blood glucose, serum lipids and antioxidant activity. The findings will contribute to the development of products with the hypolipidemic activity of chrysanthemum flavonoids and further exploration of its hypolipidemic mechanisms.

## 2. Results

### 2.1. Profiling of Phytochemical Composition and In Vitro Antioxidant of Extracts

In this study, we choose two extracts for comparison: IC-E is the crude extract of IC and IC-P is obtained by purification with macro-porous adsorption resin. In general, more than 800 metabolites were detected in IC extract, involving 66 components. There were 12 compounds with relative content of more than 1% in the two extracts, of which amino acid and derivatives, flavonoids, and carboxylic acids and derivatives were the main components (Appendix A). Compared with IC-E, the contents of flavonoids in IC-P increased (Figure 1). A total of 223 differential metabolites were screened by IC-P for the IC-E group, and the results were visualized in the form of a volcano map (Figure 2). There were 146 up-regulated differential metabolites, such as 2,3-dihydroxybenzoic acid, 3-methylindole, quercetin-3-o-sophoroside, luteolin-6-c-glucoside, etc., and 77 down-regulated differential metabolites, such as 1,2-dimethoxybenzene, 2-deoxyribose 5′-phosphate, quercetin-3-o-glucuronide, naringenin, etc. There are 62 categories of these differential metabolites and 12 representative examples are shown in Appendix A.

According to the analysis results of the extract components and the related research on chrysanthemum flavonoids reported earlier [29,30], five flavonoids were selected for quantitative determination in this study. As can be seen from Appendix A, all of the analytes showed symmetrical peak shapes, the baseline separation was obtained and the retention time and peak shapes for all of the analytes showed good correlation between the standard solution and the real sample. Correlation coefficients (R2) of regression fitting were above 0.9984 for all the analytes, indicating a good quantitative relationship between the MS responses and the analyte concentrations, which was satisfying for targeted metabolomics analysis. The recoveries determined were 93.9–115.9% for all the analytes, with all the RSDs below 1.22% (n = 3). Therefore, within the concentration range described as above, the content of the five flavonoids from high to low are cynaroside, apigenin, vitexin, homoorientin and rhoifolin, in which cynaroside and apigenin account for 33% and 5% of the IC-P.

The free radical scavenging and antioxidant activities of various IC extract have been estimated using DPPH, ABTS and HR assays (Table 1). Overall, IC extracts showed strong free radical scavenging activity against DPPH, ABTS and HR radicals. IC-P was the more effective radical scavenger with IC_50_ values of 617.5, 567.2 and 6425 trolox μg/mL FW in DPPH, ABTS and HR assays, respectively. The result suggests that IC extract after purification was more effective for extracting antioxidants with efficient free radical scavenging property.

### 2.2. Analysis of Postprandial Lipid Metabolism In Vivo

The postprandial changes in serum glucose concentration after consuming a HF meal with IC extract are shown in Figure 3A. There were significant incremental postprandial serum glucose changes in the two groups (*p* < 0.001 for time effect and time×treatment) with no statistically significant interacion of treatment effect (*p* > 0.05). The HF meal induced an increase in postprandial glucose concentration that peaked at 60 min. There were significant differences in the postprandial glucose changes consumed with the HF meal plus IC at 1, 4, 6 and 8 h time point when compared to those given a HF meal only (*p* < 0.05). In addition to the postprandial results, no significant effects were observed in the AUC of serum glucose in the two groups, but Glu-PR and maximun change were significant Appendix A.

In the two groups, significant changes in incremental postprandial serum TG were observed (*p* < 0.01 for time effect) with no statistically significant interaction of treatment and time × treatment effect (Figure 3B). Postprandial TG concentration tended to be lower at 2 and 6 h after consuming the HF meal supplemented with IC extract. However, the decrease was statistically insignificant at those time points. There was no significant difference in serum TC between the two groups (Figure 3C).

### 2.3. Antioxidant Activity In Vivo

When compared to the value at fasting state, ingestion of the HF meal resulted in a decrease in the postprandial serum T-AOC level at 0.5, 1 and 2 h (Figure 4A). At the same time point, addition of IC extract causes a significant higher postprandial T-AOC level when compared to the HF meal. The effects of IC extract on postprandial serum MDA concentration after consumption of the HF meal are presented in Figure 4B. As can be seen from the graph, HF meal intake induced a significant increase in postprandial serum MDA at 0.5 h. However, this postprandial effect was attenuated by IC extract to the HF meal (*p* < 0.01). In addition, there was no significant difference in the change of serum SOD and GSH between the two groups after the meal, and only 6h after the HF meal (Figure 4C,D).

## 3. Discussion

As an ancient flower species, chrysanthemum is common in Asian, Mediterranean and other countries. Research on the effects of chrysanthemum extract has always been a focus of attention. There is no doubt that people’s long-term habit of drinking chrysanthemum tea possesses a wide array of health potentials [31]. Numerous pharmacological studies demonstrate that chrysanthemum tea has been extensively used for health care, including anti-bacterial, anti-viral, antioxidant and anti-inflammatory capacities [32,33]. At present, chrysanthemum has been found to have anti-depressive [34], antimicrobial, [35], hepatoprotective, anti-inflammatory, antinociceptive and antiepileptic activities [36]. However, there were significant differences in composition and function among chrysanthemum cultivars [23,37]. Hodaei evaluated 17 Iranian chrysanthemum morifolium cultivars and found cultivars with high bioactive compounds [12]. Liu investigated the physicochemical characteristics, antioxidant and antiglycation activities of polysaccharides in five kinds of Chinese chrysanthemum tea and found that polysaccharides can be one of the major contributors toward their antioxidant activities [38]. Peng determined a total of 41 phenolics derived from five main species of chrysanthemum and found chlorogenic acid, 3,5-dicaffeoylquinic acid and luteolin-7-*O*-glucoside largely contributing to cellular antioxidant activity [28]. Our group’s previous research found that chrysanthemum flavonoids had a better effect on the antioxidant level and lipid metabolism-related enzyme activity [29]. Therefore, a future research direction will be to extract major compounds with specific effects from chrysanthemum for food or medical treatment. In this study, IC was selected as the research object and the metabolic component of two extracts were analyzed. Though purification, the flavonoid content in the extract of IC is increased by nearly 60% and its antioxidant activity is better. The higher content of cynaroside and apigenin in the IC-P is of great research value.

As a common chrysanthemum variety in China, IC has always been used mainly for aesthetic reasons and for eating. It is thought that brewing at high temperatures maximizes the release of the active ingredients in chrysanthemum. Modern medical studies also show that chrysanthemum can provide positive health benefits, but the effect on lipid metabolism is not clear. Animal studies have shown that chrysanthemum extract can reduce obesity in mice induced by a high-fat diet, and the extract potentially suppresses adipogenesis and lipid accumulation [11,20]. To our knowledge, the current research is the first time that investigation of the effect of water-soluble chrysanthemum extract on postprandial lipid and glucose in healthy adults has been performed. According to the repeated-measurement ANOVA, the postprandial response of lipid and glucose were significantly different between IC group and control group. Chrysanthemum extract can effectively control the rise of blood glucose and blood lipids caused by a HF diet in a short period of time, which has been confirmed in this study, but the mechanism and mode of action still need to be further explored. Our results showed that IC extracts can effectively control the rise of serum glucose caused by a HF meal in a short time. However, we also found that the serum glucose level of the IC group was higher than that of the control group 4 h after eating. This may be due to the different speed of digestion and absorption of each component of IC extract in the human body. The soluble pigments, phenols and flavonoids in chrysanthemum have low molecular weight and high solubility [39], which are released first and then quickly enter the human blood, while the polysaccharides in chrysanthemum need to be hydrolyzed into small molecules in the macromolecular state, such as monosaccharides, to be absorbed by the body [40]. Recently, it has been proven that colored fruits rich in anthocyanins and polyphenols have an effect on postprandial blood lipids [41,42], but there is no study on chrysanthemum. Therefore, the hypoglycemic effect of chrysanthemum flavonoids deserves attention. The results of this study were similar to those of Thilavech’s study [43], but since the selected subjects were from a normal population without dyslipidemia, the blood lipid level of the two groups can recover in a short time. Postprandial dyslipidemia is closely related to early atherosclerosis, mainly manifested in obstacles to of TRL clearance. Studies have shown that postprandial TG levels can represent TRLs, usually using parameters such as areas under the postprandial curve (AUC), maximum change and peak reaction. In general, in this study, IC extract mixed with HF meal has no significant difference in the postprandial lipid response between the two groups. However, IC extract can still have an impact on the human body in HF eating, mainly reflected in the maximum change and peak response Appendix A. Given that HDL-C, LDL-C, ApoA, ApoB are an acknowledged mark of lipid metabolism in the fasting state, in our research the concentration of these was also measured in the postprandial sample. However, all these markers did not show any marked change over time after meals (data not shown), which is in accordance with our previous study [44], suggesting that the postprandial rate of their catabolism and synthesis was slow. Therefore, measuring these markers will not add new information for postprandial response.

At present, available evidence demonstrates the antioxidant effect of chrysanthemum extract in vitro, cell or animal experiments, but there is little research on the human body. The few reports on the antioxidant effect of IC only focus on composition and structure, without in-depth study [22,25]. Although the current results are difficult to compare with the results of other studies due to different extraction procedures, varieties and analyzed plant parts, our observations also found that IC extracts have similar antioxidant effects to those already studied [23,24]. This study reports for the first time the antioxidant effect of IC on healthy people after a HF meal. It is well known that a large number of nutrients, such as carbohydrates and fats, are the main regulatory metabolites of dyslipidemia after a meal. In particular, the use of a HF diet will significantly increase postprandial TG and further promote the generation of free radicals. Therefore, the lipid peroxide produced by polysaturated fatty acids can form the most mutagenic by-product, MDA, through free radical chain reaction [45]. In obesity, excessive fat accumulation can promote chronic low-grade inflammation and produce a variety of proinflammatory cytokines. A chronic inflammatory process can induce the production of free radicals. Over expression of the oxidation process will damage biological molecules and reduce the activity of endogenous antioxidant enzymes, such as GSH and SOD [46]. T-AOC can reflect the antioxidant capacity of the body’s defense system to a certain extent and has a close relationship with state of health [47].

Reducing postprandial lipid level and improving antioxidant capacity have always been the goal of food nutrition intervention, with many plant compounds that can reduce lipid and improve oxidative stress in the human body still to be found. This study found that there were significant differences in antioxidant indexes between the two groups at different time points after HF eating. Therefore, it is inferred that the consumption of HF diet containing IC can enhance the antioxidant capacity of the body and this activity can maintain a high level in the blood circulation to prevent the excessive production of harmful reactive oxygen species. In addition, we found that the content of flavonoids in IC extract is relatively high. Combined with the group’s previous research on the effect of chrysanthemum flavonoids on lowering blood lipids [30], it is of practical significance to further explore the effect components of IC flavonoids on lowering blood lipids.

## 4. Materials and Methods

### 4.1. Chemicals and Reagents

All chemicals (analytical grade) used in the experiments, such as sodium carbonate, potassium persulfate and salicylic acid were purchased from Aladdin-Reagent Co., Ltd. (Beijing, China) and Sinopharm Chemical Reagent Co., Ltd., Shanghai (China), and deionized water was used. The in vitro scavenging capacity against MDA, SOD, GSH and T-AOC were determined using commercial kits (Nanjing Jiancheng Bioengineering Institute, Nanjing, China).

All the liquid high-fat loaded diet food-grade materials, casein, sucrose, lactose, maltodextrin and monoglyceride, were purchased from Henan Wanbang Industrial Co. Ltd. Butter for animal fat was purchased from Kaptein B.V. Rotterdam, Netherlands. Purified water was purchased from a Nanjing local supermarket.

### 4.2. Determination of the Total Flavone Content (TFC)

Sodium nitrite method was used to determine the rutin standard curve and TFC [48]. The sample was placed into a tube and 0.15 mL of 5% sodium nitrite solution was then added, mixed and allowed to stand at room temperature for 6 min, after which 0.15 mL of 10% aluminum nitrite solution was added, mixed and allowed to stand at room temperature for 6 min. Following the addition of 2.0 mL of 4% sodium hydroxide, 70% ethanol was added to a constant volume to 10 mL and allowed stand for 10 min. Absorbance was measured at 510 nm and the TFC was expressed as rutin equivalent/g extract.

### 4.3. Plant Materials and Extracts Preparation

Imperial Chrysanthemum is produced in Zibo, Shandong Province of China followed by 40 °C low temperature drying of the products. The dry sample (2020-c-02) was collected during December 2020 and identified by Engineering Technology Research Center of Characteristic Biological Resources. In this study the sample was ground to a powder and screened through a 120~200 mesh filter. The sample was stored at −80 °C before analysis.

For the preparation of extracts, 100 g of the powdered sample were extracted with 12 L of deionized water at 100 °C for 90 min, followed by filtration through a 0.45 μm Whatman filter paper and evaporation under reduced pressure using a rotary evaporator. After concentration, the sample (IC-E) was dried by freeze dryer (ALPHA1-2/LD-Plus, Marin Christ, German) and stored at −80 °C. The TFC was determined according to “Section 4.2”. The extraction rate and TFC of IC-E were 18.80% and 7.67 ± 0.24 mg lutin equivalent/g extract, respectively. The IC-E was redissolved by 3BV (bed volume) and vortex mixing (RCT BS25, IKA^®^-Werke GnbH&Co.KG, Staufen im Breisgau, Germany) with microporous adsorption resin (S-8) for 6 h and then filtrated. The filtered resin was added along with 2BV 70% ethanol for 6 h analysis and the same operation was performed after concentration and lyophilization to obtain the purified product (IC-P). The extraction rate and TFC of IC-P were 74.31% and 12.25 ± 0.30 mg lutin equivalent/g extract, respectively.

### 4.4. Metabolites Extraction

The IC-E and IC-P were freeze-dried. Then 20 mg aliquot of individual samples were preciselt weighed and were transferred to an Eppendorf tube, after the addition of 300 μL water (containing internal standard). After 30 s vortex, the samples were homogenized at 40 Hz for 4 min and sonicated for 5 min in an ice-water bath. The homogenization and sonication cycle were repeated 3 times. Then the samples were extracted over night at 4 °C on a shaker, then centrifuged at 12,000 rpm (RCF = 13,800 (×g), R = 8.6 cm) for 15 min at 4 °C. The supernatant was carefully filtered through a 0.22 μm microporous membrane, then the resulting supernatants were diluted 50 times with methanol/ water mixture (*v*:*v* = 3:1, containing internal standard), vortexed for 30 s and transferred to 2 mL glass vials, when 40 μL was taken from each sample and pooled as QC samples, stored at −80 °C until the UHPLC-MS analysis.

### 4.5. Profiling of Phytochemical Composition by UHPLC-MS

The UHPLC (ExionLC AD, AB SCIEX, Framingham, MA, USA) separation was carried out using an EXIONLC System (Sciex). The mobile phase A was 0.1% formic acid in water and the mobile phase B was acetonitrile. The column temperature was set at 40 °C. The auto-sampler temperature was set at 4 °C and the injection volume was 2 μL. A Sciex QT 6500 + (Sciex Technologies, AB SCIEX, Framingham, MA, USA), was applied for assay development. Typical ion source parameters were: IonSpray Voltage: +5500/−4500 V, Curtain Gas: 35 psi, Temperature: 400 °C, Ion Source Gas 1:60 psi, Ion Source Gas 2:60 psi, DP: ±100 V [49].

### 4.6. The Main Flavonoid Compounds Analysis

The HPLC equipment and ion source parameters refer to “Section 4.5”. The MRM parameters for each of the targeted analytes were optimized using flow injection analysis, by injecting the standard solutions of the individual analytes into the API source of the mass spectrometer (QT 6500+, AB SCIEX, Framingham, MA, USA). Several of the most sensitive transitions were used in the MRM scan mode to optimize the collision energy for each Q1/Q3 pair. Among the optimized MRM transitions per analyte, the Q1/Q3 pairs that showed the highest sensitivity and selectivity were selected as ‘quantifier’ for quantitative monitoring. The additional transitions acted as ‘qualifier’ for the purpose of verifying the identity of the target analytes.

### 4.7. Antioxidant Activity

#### 4.7.1. DPPH Radical Scavenging Activity

The DPPH scavenging activity was evaluated according to Gong’s method [50] with slight modification. The reaction was performed using 0.5 mL samples at various concentration (from 0.01 to 0.5 mg of dry weight per mL) and 2.1 mL of 0.1 mM DPPH methanol solution. The mixture was vigorously vortexed and allowed to stand for 60 min in the dark. The absorbance was measured at 517 nm and Trolox was used as t_h_e positive standard. The percent rate of DPPH free radicals was calculated as follow:(1)$$\%{\mathrm{DPPH}}^{\cdot }\mathrm{scavenging}\;\mathrm{rate}=\left[1-\frac{A_{sample}}{A_{blank}}\right]\times 100$$
where *A_blank_* is the absorbance of the blank and *A_sampl_*_e_ is the absorbance of the sample.

#### 4.7.2. ABTS^+^ Radical Scavenging Activity

The determination of ABTS scavenging rate was based on the method used by Chen [51]. ABTS^+^ cation radical was generated by oxidation of 7.4 mM ABTS stock solution with 2.6 mmol/L potassium persulfate and the working solution was kept in the dark for 12–16 h. Then ABTS^+^ working solution was diluted with PBS (5 mM phosphate buffered saline, pH 7.4) to obtain an absorbance of 0.70 ± 0.02 at 734 nm. Subsequently, a 1 mL sample at various concentration (from 0.1 to 1 mg of dry weight per mL) was mixed with 3 mL ABTS^+^ and the reaction mixture was allowed to stand for 6 min at room temperature in the dark. Then the absorbance was measured at 734 nm and the ABTS^+^ radical scavenging activity determined in terms calculated as previously described for the DPPH assay. In both assays, the IC_50_ values which represent the concentration of sample required for the inhibition of 50% DPPH or ABTS radicals was determined, where *A_sample_* was the absorbance of sample, *A_control_* was the absorbance of H_2_O_2_ replaced distilled water and *A_blank_* was the absorbance of the blank.

#### 4.7.3. Hydroxyl Radical (HR) Scavenging Activity

HR scavenging rate was generated by the salicylic acid method with slight modification [52]. The reaction mixture contained 1 mL of ferrous sulfate (9 mM), 1 mL of salicylic acid–ethanol (9 mM) and 1 mL of sample at various concentrations (from 1 to 10 mg of dry weight per mL). The reaction was started by 1 mL of hydrogen peroxide (8.8 mM). After incubating at 37 °C for 0.5 h, the absorbance of samples was measured at 510 nm and the results were expressed as IC_50_ Trolox μg/mL FW. The hydroxyl radical scavenging activity was calculated by the following formula:(2)$$\%\mathrm{HR}\;\mathrm{scavenging}\;\mathrm{rate}=\left[1-\frac{A_{sample}-A_{control}}{A_{blank}}\right]\times 100\%\;$$
where *A_sample_* was the absorbance of sample, *A_contro_*_l_ was the absorbance of H_2_O_2_ replaced distilled water and *A_blank_* was the absorbance of the blank.

### 4.8. Postprandial Lipid Metabolism Activity

#### 4.8.1. Postprandial Lipid Metabolism Activity

In order to avoid the difference of metabolism caused by different chewing degree and eating speed of different subjects after chewing food in most domestic post-prandial lipid metabolism experiments, an edible liquid high-fat loaded diet, which can be directly drunk, was adopted in this study. According to previous research [46], the supply of high-fat load diet in this project was based on the body weight (bw) of the participant at the rate of 18 kcal/kg∙bw, in which fat is 50% of total energy (butter), carbohydrate ratio is 35% (sucrose 63%, maltodextrin 30%, lactose 7%), and protein 15% (casein). High-fat load diet ingredients (60 kg/person): casein 40.5 g, sucrose 59.5 g, lactose 6.6 g, buffer 60 g, maltodextrin 28.4 g, emulsifier 0.6 g monoglyceride were added and the volume was constant to 300 mL. The production process is as follows: mixing raw materials with water-heating-shearing (80 °C, 2000 r/min)—homogenization–sealing and packing—sterilization.

#### 4.8.2. Study Subject

The sample was composed of 37 adults recruited from the local community. They were classified into two groups based on the fasting blood TG level: Imperial chrysanthemum group (IC group, n = 19) and healthy control group (n = 18). Subjects were excluded if they had diabetes, coronary heart disease, hypertension, gastrointestinal diseases and other chronic inflammatory diseases. Subjects were included following: age between 20 and 40 healthy Asian adults, body mass index (BMI) between 18.5 and 23.9 kg/m^2^, Triglyceride (TG) < 1.7 mmol/L, total cholesterol (TC) < 5.72 mmol/L, high density lipoprotein cholesterol (HDL-C) > 0.91 mmol/L and low-density lipoprotein cholesterol (LDL-C) < 3.6 mmol/L. Subjects were also required to be nonsmokers, nonvegetarian, non-drinkers and nonusers of dietary supplements or gastrointestinal medications. The baseline characteristics of subjects are summarized in Table 2.

#### 4.8.3. Study Design

The study used a randomized controlled design. The subjects maintained normal lifestyle and dietary habits before the experiment and on the day prior to the test day. Subjects were asked to refrain from alcohol, high-fat food and staying up late. After a 12-h overnight fast, an intravenous catheter was inserted into a forearm vein for collecting blood samples. After taking a fasting blood sample, each participant was requested to consume one of the test diets. Subsequent blood samples were collected at 0.5, 1, 2, 4, 6 and 8 h after diet consumption, The serum was separated by centrifugation at 4000× *g* for 15 min then analyzed immediately.

#### 4.8.4. Laboratory Assessments

Concentration of glucose, TG, TC, HDL-C, LDL-C, ApoA, ApoB, MAD, SOD, GSH and T-AOC were determined in serum samples from T = 0, 0.5, 1, 2, 4, 6, 8 h after meal consumption. TG, TC, HDL-C and LDL-C concentrations were determined by enzymatic assays. Glucose concentrations were determined by the glucose oxidase method. The concentrations of ApoA and ApoB were detected by immune turbidimetric method. Concentrations of MAD, SOD, GSH and T-AOC were measured using kits.

### 4.9. Statistical Analysis

Data were expressed as mean ± SD for normally distributed. The incremental areas under the postprandial curve (iAUC) or the decremental AUC (diAUC) and maximal change were used to evaluate the overall response during postprandial period. AUC was calculated using GraphPad Prism (version8.0.1, GraphPad Software, Inc., La Jolla, CA, USA). Maximal change was calculated by subtracting fasting concentrations from maximal value or by subtracting minimal value from fasting concentrations. Analysis of variance (ANOVA) for repeated measures was used to analyze the time and meals interaction within subject groups and the time and group interaction within meals. SCIEX Analyst Work Station Software (Version 1.6.3, AB SCIEX, Framingham, MA, USA) was employed for MRM data acquisition and processing. MS raw data (wiff) files were converted to the TXT format using MSconventer. In-house R program and database were applied to peak detection and annotation. All these analyses were performed using SPSS (version 22.0, Chicago, IL, USA). Values with *p* < 0.05 were considered statistically significant in all cases.

## 5. Conclusions

In this study, the metabolites of two extracts of IC were analyzed and compared. It was found that the content of flavonoids in the ICP was increased and cynaroside and apigenin were the main components. In addition, this is the first study to demonstrate that acute consumption of an HF meal accompanied with IC extract decreases postprandial serum glucose, and IC extracts significantly improve serum antioxidant status responses to the HF meal by increasing serum T-AOC. However, IC could not reduce the effect of HF meal-induced in postprandial TG and TC. The results show an important role played in maintaining glucose metabolism and a certain protective effect on human metabolism after HF diet. IC has potential application value in the pharmaceutical and food industries. Importantly, the study might contribute to the development of functional foods and pave the way for the hypoglycemic drugs of IC as consolidated source of bioactive ingredients.

## 6. Patents

The work reported in this manuscript has obtain an authorized patent from the People’s Republic of China (ZL 2021 2 0817783.8).

## Figures and Tables

**Figure 1 molecules-28-00579-f001:**
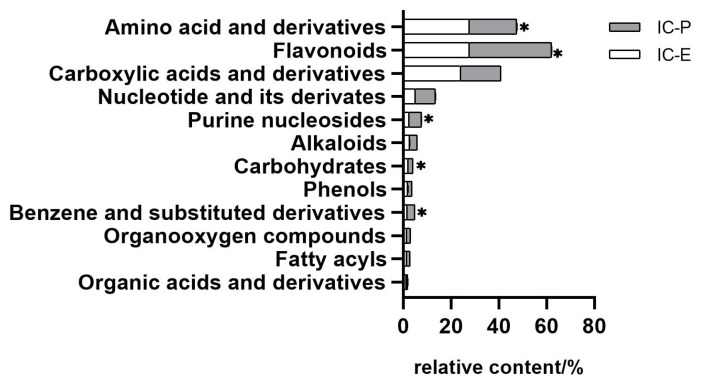
Comparison of relative contents of different components in extracts. Bar with * are significantly (*p* < 0.05) different.

**Figure 2 molecules-28-00579-f002:**
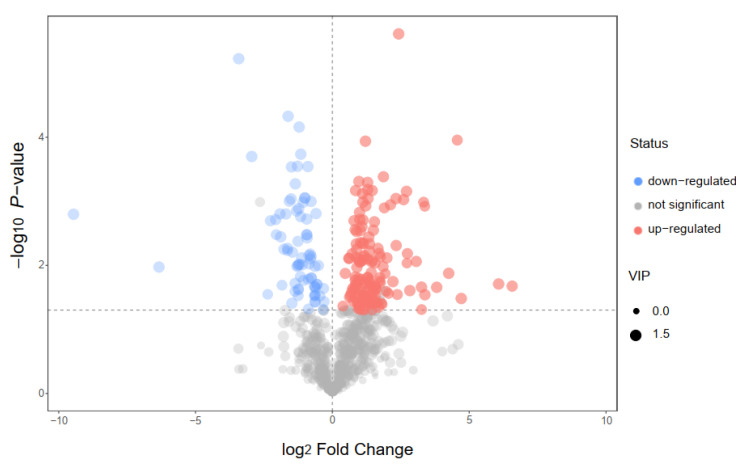
Volcano plot for group IC-P vs. IC-E.

**Figure 3 molecules-28-00579-f003:**
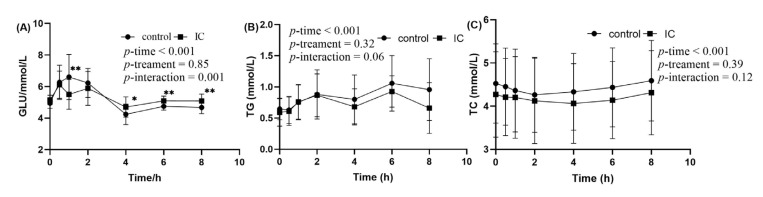
The concentration of glucose, TG and TC in serum sample within 8 h after meal. (**A**) postprandial serum glucose concentration; (**B**) postprandial serum TG concentration; (**C**) postprandial serum TC concentration. Data are expressed as mean ± SD (n = 18), with *: *p* < 0.05, **: *p* < 0.01.

**Figure 4 molecules-28-00579-f004:**
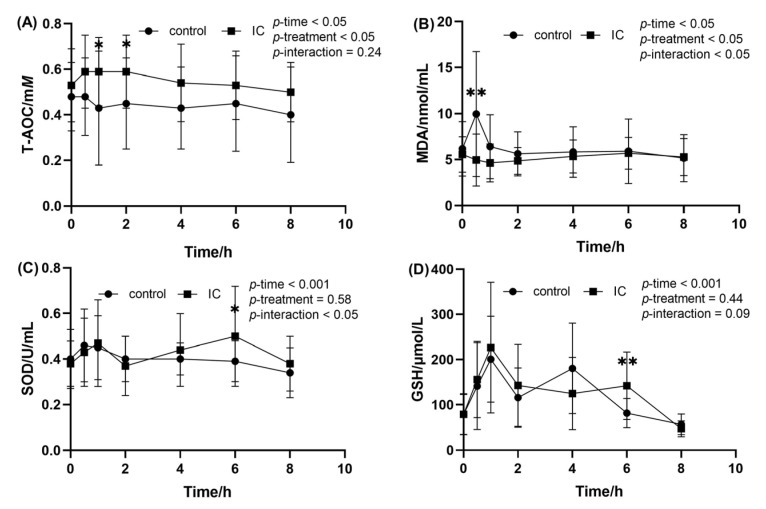
The concentration of MDA, SOD, GSH and T-AOC in the serum sample within 8 h after meal. (**A**) postprandial serum T-AOC concentration; (**B**) postprandial serum MDA concentration; (**C**) postprandial serum SOD concentration; (**D**) postprandial serum GSH concentration. Data are expressed as mean ± SD (n = 18), with *: *p* < 0.05, **: *p* < 0.01.

**Table 1 molecules-28-00579-t001:** In vitro antioxidant of IC extracts.

Parameters	IC-E	IC-P	*p*
DPPH (IC_50_ trolox μg/mL FW)	939.6 ± 1.93	617.5 ± 1.46	<0.05
ABTS (IC_50_ trolox μg/mL FW)	588.6 ± 2.44	567.2 ± 2.94	
HR (IC_50_ trolox μg/mL FW)	6697 ± 8.60	6425 ± 4.63	

Values are mean ± SD (n = 3).

**Table 2 molecules-28-00579-t002:** Subjects’ baseline information on postprandial lipid metabolism.

	Control Group (n = 18)	IC Group (n = 19)	*p*
Sex (male/female)	6/12	7/12	NS
Age	25.22 ± 2.44	25.26 ± 2.75	NS
Height/m	165.08 ± 6.602	166.21 ± 7.729	NS
Weight/kg	56.43 ± 6.690	58.82 ± 7.434	NS
BMI/kg/m^2^	20.65 ± 1.471	21.09 ± 1.506	NS
TC/mmol/L	4.52 ± 0.844	4.30 ± 0.878	NS
TG/mmol/L	0.70 ± 0.230	0.75 ± 0.227	NS
HDL-C/mmol/L	1.61 ± 0.300	1.51 ± 0.250	NS
LDL-C/mmol/L	2.29 ± 0.560	2.25 ± 0.642	NS

Data are means ± S.E.M.

## Data Availability

Not applicable.

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
