# Peer review of "Analysis of Chemical Constituents of Chrysanthemum morifolium Extract and Its Effect on Postprandial Lipid Metabolism in Healthy Adults"

_molecules, 2023, doi:10.3390/molecules28020579_

Round 1
Reviewer 1 Report
The author selected imperial chrysanthemum (IC) which is a traditional Chinese medicinal herb as the main research object, analyze the composition of extract and determine the effect on changes in postprandial glycemic and lipemic response and antioxidant status in adults after consumption of a high-fat (HF) meal. The finding suggests that IC can be used as a natural ingredient for reducing postprandial lipemia and improving the antioxidant status after consuming HF meal.
This current study has done a lot of work in the composition and basic application of imperial chrysanthemum deionized water extract. It's an interesting job with some potential in pharmaceutical applications. The scientific work had been clearly explained with well-organized figures and tables. Therefore, I recommended this work can be accepted after several corrections need to be made listed as follows:
1. The title seems a little long. Could the author simplify it to make it more consistent with the research order in this manuscript and attract readers?
2. For the Keywords, the author should choose more suitable keywords to increase the search.
3. For the Introduction, it's better to write a another paragraph for "This article mainly studied…"
4. For Figure 2, the font size of the illustration should be larger.
5. For Figure 3, please explain the symbol of significant difference in the figure note.
6. In the “2.3. Antioxidant activity in vivo” section, Figure 1 should be Figure 4.
7. In the 4.6. Antioxidant activity, the calculation formula should have a separate row
8. In Table S1, the numbers in chemical formulas should be subscript.
9. This manuscript involves human experiments, it is suggested to attach the ethical proof in the attachment.
10. For references, some references are old, and authors should try to find newer ones.
11. The manuscript has some typographical and grammatical errors which must be corrected.
Author Response
Manuscript ID: molecules-2114124
TITLE: Effects of chrysanthemum extract on postprandial lipid metabolism in healthy adults and analysis of its extract’s
Analysis of chemical constituents of Imperial Chrysanthemum extract and its effect on postprandial lipid metabolism in healthy adults
Dear editor,
Thank you very much for your attention and the referee’s comments on our manuscript. These comments have been the most helpful in revision of our manuscript, and we have modified the manuscript accordingly and all the changes we have made were highlighted in yellow in the revised manuscript. Answers to reviewers’ questions are listed below point by point. Finally, we wish to make this opportunity to thank you for your consideration of our paper for publication in the journal Molecules. Please see the attachment.
Responses to comments of reviewers
We are very grateful to your comments for the manuscript. According to your advice, we revised the relevant part in manuscript, and the questions were answered below.
Point 1: The title seems a little long. Could the author simplify it to make it more consistent with the research order in this manuscript and attract readers?
Response 1: Thanks for the reviewer’s kind comments.
We have adjusted the topic to make it more consistent with the research order in the manuscript.
Point 2: For the Keywords, the author should choose more suitable keywords to increase the search.
Response 2: Thanks for the reviewer’s kind comments.
We have modified and added keywords, “Imperial Chrysanthemum, UHPLC-MS, flavonoids, high-fat meal, postprandial lipemia, antioxidant status”.
Point 3: For the Introduction, it's better to write another paragraph for "This article mainly studied…"
Response 3: Thanks for the reviewer’s kind comments.
In the introduction, at the end of the second paragraph, we have a description of the purpose of the experiment. Your suggestion is still important, so we have made adjustments.
Point 4: “For Figure 2, the font size of the illustration should be larger.” 、 “For Figure 3, please explain the symbol of significant difference in the figure note.” 、 “In the “2.3. Antioxidant activity in vivo” section, Figure 1 should be Figure 4.” 、“In the 4.6. Antioxidant activity, the calculation formula should have a separate row.” 、“In Table S1, the numbers in chemical formulas should be subscript.” 、“For references, some references are old, and authors should try to find newer ones.” 、“The manuscript has some typographical and grammatical errors which must be corrected.”
Response 4: Thanks for the reviewer’s kind comments.
The above suggestions have been revised in the article.
Point 5: This manuscript involves human experiments, it is suggested to attach the ethical proof in the attachment.
Response 5: Thanks for the reviewer’s kind comments.
We have attached the ethical certificate in the attachment.
Finally, thank you again for your careful review of the paper, which provided me with very valuable advice. I wish you good health and smooth work!
Sincerely, Chen
2022/12/29
Reviewer 2 Report
I suggest that the article titled “Effects of Chrysanthemum Extract on Postprandial Lipid Metabolism in Healthy Adults and Analysis of Its Extract’s Chemical Constituents” be accepted for publication after the revisions suggested below are carried out:
- The scientific name of the species must appear in the title.
- Were the authors able to carry out the experiments with only 1 g of dry material? Describe the extract masses for each step in items 4.2 and 4.3.
- The authors do not describe the name of the species in item 4.2.
- At no point in the abstract, nor in the results and discussion, is the difference between the extracts named IC-P and IC-E made clear. The description of how these extracts were obtained is reported only in the experimental section.
- In table S2 the column named "extract mass" is wrong. In this case it would be Molecular mass or Molecular ion.
- To annotated (and not identify) the molecules, the authors must present the mass results in high resolution and add a column with the relative error calculation.
Author Response
Manuscript ID: molecules-2114124
TITLE: Effects of chrysanthemum extract on postprandial lipid metabolism in healthy adults and analysis of its extract’s
Analysis of chemical constituents of Imperial Chrysanthemum extract and its effect on postprandial lipid metabolism in healthy adults
Dear editor,
Thank you very much for your attention and the referee’s comments on our manuscript. These comments have been the most helpful in revision of our manuscript, and we have modified the manuscript accordingly and all the changes we have made were highlighted in yellow in the revised manuscript. Answers to reviewers’ questions are listed below point by point. Finally, we wish to make this opportunity to thank you for your consideration of our paper for publication in the journal Molecules.Please see the attachment.
Responses to comments of reviewers
We are very grateful to your comments for the manuscript. According to your advice, we revised the relevant part in manuscript, and the questions were answered below.
Point 1: The scientific name of the species must appear in the title.
Response 1: Thanks for the reviewer’s kind comments.
We have supplemented and modified the title.
Point 2: Were the authors able to carry out the experiments with only 1 g of dry material? Describe the extract masses for each step-in item 4.2 and 4.3.
Response 2: Thanks for the reviewer’s kind comments.
We actually used 100g sample for extraction. Considering the visual representation of the extraction ratio, we wrote the dosage of 1g chrysanthemum. Your question made us realize that this would lead to misunderstanding, so we improved the description of experimental methods and supplemented the extract mass of each step.
Point 3: At no point in the abstract, nor in the results and discussion, is the difference between the extracts named IC-P and IC-E made clear. The description of how these extracts were obtained is reported only in the experimental section.
Response 3: Thanks for the reviewer’s kind comments.
IC-E is the crude extract of Imperial Chrysanthemum, and IC-P is the product after further purification on the basis of IC-E. We have supplemented the extraction rate and flavonoid content of those two extracts in the method, and compared the extracts in ‘2.1’. This article mainly focuses on the content, types and antioxidant effects of flavonoids from the two extracts. We found that the flavonoid content of the purified chrysanthemum extracts increased and its antioxidant activity increased, providing a reference for future human studies.
Point 4: In table S2 the column named "extract mass" is wrong. In this case it would be Molecular mass or Molecular ion.
- To annotated (and not identify) the molecules, the authors must present the mass results in high resolution and add a column with the relative error calculation.
Response 4: Thanks for the reviewer’s kind comments.
The above suggestions have been revised in the “table S2”.
Finally, thank you again for your careful review of the paper, which provided me with very valuable advice. I wish you good health and smooth work!
Sincerely, Chen
2022/12/29
Round 2
Reviewer 2 Report
Dear Authors,
I suggest that the following suggestions be carried out so that the article is accepted for publication.
- Imperial Chrysanthemum is not the scientific name of the species
- In order to be sure of the annotated compounds, the mass values Q1 and Q3 must be in high-resolution.
Author Response
Manuscript ID: molecules-2114124
TITLE: Effects of chrysanthemum extract on postprandial lipid metabolism in healthy adults and analysis of its extract’s
Analysis of chemical constituents of Chrysanthemum Morifolium extract and its effect on postprandial lipid metabolism in healthy adults
Dear editor,
Thank you very much for your attention and the referee’s comments on our manuscript. These comments have been the most helpful in revision of our manuscript, and we have modified the manuscript accordingly and all the changes we have made were highlighted in yellow in the revised manuscript. Answers to reviewers’ questions are listed below point by point. Finally, we wish to make this opportunity to thank you for your consideration of our paper for publication in the journal Molecules.
Responses to comments of reviewers
We are very grateful to your comments for the manuscript. According to your advice, we revised the relevant part in manuscript, and the questions were answered below.
Point 1: Imperial Chrysanthemum is not the scientific name of the species
Response 1: Thanks for the reviewer’s kind comments.
We have modified the title.
Point 2: In order to be sure of the annotated compounds, the mass values Q1 and Q3 must be in high-resolution.
Response 2: Thanks for the reviewer’s kind comments.
Your suggestion is very good. Unfortunately, our unit does not have a mass spectrometer with higher resolution. In addition, we will focus on the flavonoid in chrysanthemum extract.
Finally, thank you again for your careful review of the paper, which provided me with very valuable advice. I wish you good health and smooth work!
Sincerely, Chen
2022/12/30
